# In Situ Calibration of Wetlabs Chlorophyll Sensors: A Methodology Adapted to Profile Measurements

**DOI:** 10.3390/s23052825

**Published:** 2023-03-04

**Authors:** Joëlle Salaün, Marc Le Menn

**Affiliations:** Shom, French Navy Hydrographic and Oceanographic Service, 29228 Brest, France

**Keywords:** calibration, chlorophyll sensor, in situ measurements, HPLC, Wetlabs

## Abstract

Measurement of chlorophyll *a* content in the ocean is essential for biomass assessment, finding the optical properties of seawater, and calibration of satellite remote sensing. The instruments used for this purpose are mostly fluorescence sensors. The calibration of these sensors becomes a crucial point to ensure the reliability and quality of the data produced. The technology of these sensors is based on the principle that a concentration of chlorophyll *a* in µg per liter can be calculated from an in situ fluorescence measurement. However, the study of the phenomenon of photosynthesis and cell physiology teaches us that the yield of fluorescence depends on many factors that are difficult or impossible to reconstitute in a metrology laboratory. This is the case, for example, of the algal species, its physiological state, the greater or lesser presence of dissolved organic matter in the water, the turbidity of the environment, or the surface illumination. What approach should be adopted in this context to achieve a better quality of the measurements? This is the objective of the work we present here, which is the result of nearly ten years of experimentation and testing to optimize the metrological quality of chlorophyll *a* profile measurement. The results we obtained allowed us to calibrate these instruments with an uncertainty of 0.2–0.3 on the correction factor, with correlation coefficients higher than 0.95 between the sensor values and the reference value.

## 1. Introduction

The measurement of chlorophyll *a* in vivo with fluorescence sensors is widely used. The metrological verification of these instruments before deployment is essential in order to ensure the good quality of measurements.

The purpose of this publication is to examine ten years of in situ fluorometer data and compare them to HPLC analyses performed on samples collected in parallel, in order to propose a method to control the calibration of these instruments.

The principle of this measurement is based on the fluorescence property of the molecule. When a solution of chlorophyll *a* is illuminated by a light beam with a wavelength of 432 nm, it changes from its ground state, s0, to a more energetic state, s1′. Then, it relaxes in two phases, from s1′ to s1, by dissipating a small amount of heat, and finally from s1 to s0, by emitting a less energetic light beam at around 670 nm.

In the living cell, chlorophyll has a third mode of relaxation, by producing electrons whose energy is used to allow the synthesis of carbohydrates, lipids, and proteins. This is the principle of photosynthesis.

In vivo, these three modes of relaxation are in competition and will vary according to the physiological state of the cells.

There is a competition between photochemistry, heat dissipation, and fluorescence. When one of the pathways predominates, the other two decrease accordingly. The share of each of these relaxation pathways in vivo varies from 3% to 5% for fluorescence, 0–20% for photochemistry, and 75–97% for heat dissipation [1].

Maximum fluorescence yield is obtained when the photochemistry is minimal, which often corresponds to a state of physiological stress caused by the lack of nutrients.

Moreover, within the living cell, pigments are frequently bound to other compounds such as proteins, which significantly modifies their spectral characteristics.

In 2000, Poryvkina et al. [2] studied the variation of the emission intensity fluorescence at 680 nm in vivo for 15 algae cultures subjected to an excitation wavelength ranging from 400 to 650 nm. They showed that the in vivo absorption maxima of chlorophylls a, b, and c are shifted by about 15 nm depending on the species of algae, while the absorption maximum of carotenoids could be increased by 90 nm.

In 2001, Vivian Lutz [3] also showed that the absorption and fluorescence spectra are modified in vivo depending on the algal species and the luminosity.

In situ fluorometers generate an excitation at around 430 nm and collect the fluorescence emission of chlorophyll *a* contained in the cells at around 670 nm.

To determine a concentration of chlorophyll, it is necessary to know the specific absorption coefficient of chlorophyll, as well as its fluorescence quantum yield.

During calibration by the manufacturer, these two terms are calculated globally by immersing the instruments in a culture of “mono-specific” algae, under controlled environmental conditions (light, temperature, nutrients) and during the exponential growth phase. In this way, it is assumed that the specific absorption coefficient and the fluorescence quantum yield are constant, independently of the algal species, the growth phase, and the environmental conditions.

However, the specific absorption coefficient is not constant in the intracellular environment and many interferences can modify the fluorescence yield of in vivo measurement instruments.

This is particularly the case of the lighting conditions at the time of the measurement or “non-photochemical quenching”, i.e., the particularity of cells undergoing very strong lighting, particularly at the subsurface, to protect themselves from premature aging due to UV by inhibiting transient photosynthesis [4,5,6].

The more or less significant presence of CDOM (colored dissolved organic matter) may overestimate the result, while the turbidity of the medium can affect the correct reception of the fluorescence intensity, underestimating the amount of chlorophyll *a*.

All these factors make it impossible to establish a single correction coefficient for the same sensor deployed over a large geographical area and over a long period. In other words, the same instrument cannot be corrected in the same way if it is used in the North Atlantic or in the Mediterranean, in winter or in spring.

Better correlation with laboratory analyses is frequently found at a specific geographic point, traversing the photic layer over a 3–4 min period using a CTD (conductivity, temperature, depth) profiler at a speed of 1 m per second. A greater homogeneity of the algal population is observed along a vertical profile.

Since a robust calibration coefficient cannot be established prior to field use, data collected in real time are biased in the absence of concurrent sampling.

The aim of this publication is to show that calibration of chlorophyll sensors with dyes or algal cultures prior to deployment induces a significant bias in the field data. Only in situ calibration with sea samples can provide unbiased quantitative data.

Based on this observation, we adopted a procedure explained in this publication. It consists in quantifying fluorescence data after in situ calibration. Before deployment, we ensure that the sensor is functioning properly, what relative calibration values can be used during the campaign, and finally, how to optimize the post-campaign calibration method thanks to the seawater samples.

## 2. State-of-the-Art

Most manufacturers of in situ fluorometers calibrate their instruments at two points: “zero”, obtained by masking the optical faces with black tape, and maximum output by exposing it to a solid fluorophore (fluorescence stick) or liquid (crushed spinach leaf, Coca-Cola^®^, or Sprite^®^, etc.). Chelsea describes a detailed method for calibrating its Aquatracka3 with chlorophyll solutions in acetone. However, user experience has prompted some manufacturers such as Seabird to specify in most of their literature that only in situ calibration with concomitant water sampling can approximate the true chlorophyll value in µg/L. Excerpts from the Seabird application note No. 41 (https://www.seabird.com/application-notes (accessed on 15 December 2022) state that:

“While the factory-supplied scale factor can be used to obtain approximate values, field calibration is highly recommended.

For example, the relationship between fluorescence and chlorophyll *a* is highly variable and is not easy to determine in the laboratory. Species distribution, ambient light level, and health of the stock are just some of the factors that affect the relationship. To accurately measure chlorophyll-a concentration with a fluorometer, perform calibrations on seawater samples with concentrations of plankton populations that are similar to what is expected in situ. Determine chlorophyll-a concentrations independently, and use those concentrations, as well as readings from the fluorometer, to determine the correct scale factor. The scale factor is correct as long as the condition of the plankton population does not change; the condition does change with season and geographic location.”

In 2011, Alan Earp [7] published an article where he studies various fluorescent substances in aqueous solution with the aim of calibrating laboratory instruments. These substances are: fluorescein, Basic Blue 3, and rhodamine WT (Figure 1).

With these three compounds, it is possible to show the linearity of the response of an instrument. Fluorescein is the compound whose absorption maximum is close to that of chlorophyll, but its cost is significantly higher than rhodamine.

In 2011, Collins Roesler [8] presented an approach to calibrate and correct the measurements of fluorimeters fitted to fixed buoys in the Gulf of Maine. The optical instrumentation consisted of Wetlabs chlorophyll *a*/turbidity ECO-series FLNTU sensors.

The sensors are calibrated by the manufacturer before and after each deployment and in the laboratory using diatom cultures. The drift of the sensors is verified by measuring the dark count before and after each deployment. Biofouling is quantified on two sensors by measuring the algal culture in the laboratory on the last day of sensor1 deployment and measuring the same culture on the first day of sensor2 deployment. The difference between the two sensors is a correction factor that is applied a posteriori to sensor1, with the drift being assumed to be linear over time.

This strategy is suitable for moorings, but in our case, when measuring in-profile, our instruments were not subjected to biofouling. On the other hand, we have the possibility of going further by taking samples during the measurements.

In 2011, Xiaogang Xing [9] attempted to model the relationship between irradiance and fluorescence from profiles obtained in various ocean regions by eight Bio-Argo floats.

This resulted in the following equation:(1)$$\ln Ed\left(\lambda ,z\right)+\overset{n}{\underset{1}{\sum }}Kw\left(\lambda \right)\Delta z=\ln Ed\left(\lambda ,0-\right)-F^{e\left(\lambda \right)}\overset{n}{\underset{1}{\sum }}\left[\chi \left(\lambda \right)fluo{\left(z\right)}^{e\left(\lambda \right)}\right]\Delta z$$
where *Ed*(λ, *z*) is the irradiance at the wavelength λ in nm and depth *z*, *Ed*(λ,0−) is the irradiance at λ (nm) at the surface, *Kw*(λ) ∆z is the diffuse attenuation generated by pure seawater at λ nm on the water column, *F* is the desired correction factor, *χ*(λ) is the coefficient due to the part of the attenuation caused by the biological material, and fluo (*z*) is the fluorescence measured by the sensor.

This equation makes it possible to estimate the non-photochemical quenching (NPQ). It addresses the correction of measurements in the euphotic layer only with the assumption that the profile of chlorophyll *a* as well as the decrease in irradiance are uniform in this euphotic layer. In particular, the estimate does not consider the variability of the fluorescence yield depending on the species of algae, their physiological state, and the presence of CDOM.

In 2016, Emmanuel Boss [10] proposed an approach to correct the in situ fluorescence values of the data from the fluorimeters fitted to the floats launched as part of the SOCCOM project (Southern Ocean Carbon and Climate Observations and Modeling). He was particularly interested in the corrections to be made on the profiles to take account of non-photochemical quenching (NPQ).

Prior to deployment, each float is attached to the carousel of a CTD with Niskin bottles. Its calibration is performed with reference to laboratory HPLC values, and this calibration is assumed to be constant thereafter.

On each profile, an in situ “dynamic” dark value is calculated, corresponding to the median value of the 10 lowest values of the profile, and this first correction is applied to the values of the sensor.

For floats equipped with a radiometer (PAR), the NPQ correction is applied at each depth where the PAR value is greater than 80 Einstein m^−2^·s^−1^ (mol equivalent m^−2^·s^−1^). This correction is obtained using the method of Sackman et al. [11], which calculates the ratio of deep fluorescence to backscatter at 700 nm (Rf/b700) for the portion of the profile below the mixing layer and extrapolates the surface fluorescence profile assuming that the ratio is constant throughout the water column, which gives:non-quenched fluorescence = surface backscattering * Rf/b700(2)

The correction is also calculated according to the method of Xing et al. (2011) described above, which uses Equation (1).

Finally, the average is established between the two methods in order to determine the NPQ correction.

In addition to the fact that this method requires simultaneous measurements of irradiance and fluorescence, it does not consider the diversity of fluorescence yields according to the species of phytoplankton and the geographical areas through which the floats will subsequently pass. A primary calibration of the floats, made with reference to a single-point CTD measurement, cannot completely solve this problem.

In 2017, Xiaogang Xing [12] estimated the correction of fluorescence measurements in the presence of CDOM by examining profiles obtained in the Black Sea and South Pacific. It appears that in this area, the fluorescence measurement of the chlorophyll *a* sensor increases monotonically below the chlorophyll minimum. His hypothesis is that this interference can only be due to the presence of CDOM.

He derives the following relationship:Chla_sensor_ corrected = Chla_sensor_ − Chla_dark_ − CDOM_interference_(3)
where Chla_sensor_ is the fluorescence raw value in volts measured by the chlorophyll sensor, Chla_dark_ is the raw value in volts when a black tape is used to mask the optical window of the chlorophyll sensor, and CDOM_interference_ is the contribution in volts of the CDOM to the measurement made by the chlorophyll sensor.

In order to calculate the interference of the CDOM, the profiles are cut in half at 200 m, believing that below this depth, only the CDOM can contribute to the response of the chlorophyll sensor. The raw volt values, corrected by the dark count, of both sensors (chlorophyll and CDOM) are collected from 200 m to the bottom, in order to plot the regression graph Chla_bottom_~CDOM_bottom_:Chla_bottom_ = (FC ∗ CDOM_bottom_) + C(4)
where FC is the slope of the regression line and C is the offset.

Then, the correction is applied to the profile of the surface at 200 m:Chla_corrected_ = Chla_surface_ − ((CDOM_surface_ ∗ FC) + C)(5)

In 2017, Collin Roesler [13] published in the journal “Limnology and Oceanography: Methods”, an article entitled “Recommendations for obtaining unbiased chlorophyll estimates from in situ chlorophyll fluorometers: A global analysis of Wetlabs ECO sensors”.

He showed that the Wetlabs ECO-FLs exhibit a specific bias, independent of all other known biases, but also that optical sensors target the fluorescence produced in photosystem II, while chlorophyll is found in other places.

The fluorescence intensity Fchl formula is similar to the Beer–Lambert’s law for spectroscopy:(6)$$F_{chl}=\phi_{F}\int_{400}^{750}E\left(\lambda \right)\times a_{chla}^{*}\left(\lambda \right)\times chla$$ where $\phi_{F}$ is the fluorescence yield, *E*(λ) is the photosynthetically active radiation for a wavelength λ, i.e., “sensor excitation irradiance” expressed in µmol.m^−2^.s^−1^.nm^−1^, $a_{chla\;}^{*}\left(\lambda \right)$ is the specific absorption coefficient of chlorophyll *a* at a wavelength λ in m^2^.mg^−1^, and chla is the amount of chlorophyll *a* in mg.

The formula for converting fluorescence into the chlorophyll *a* concentration (mg/m^3^) is derived from Equation (6):(7)[chla]=Fchl ϕF∫400750E(λ)×achla*(λ)

This is the formula used by the optical chlorophyll sensors. However, in reality, the terms of the equation are not measured quantitatively because:-Only a fraction of chlorophyll *a* is detected.-The specific absorption coefficient of chlorophyll *a* (*a^*^_chla_* (λ)) is not known when chlorophyll is bound to proteins inside the living cell.-The fluorescence yield, *ϕ_F_*, varies depending on the algal population and its physiological state.

In practice, globally, sensor manufacturers calculate the term $\;[\;\phi_{F}\int_{400}^{750}E\left(\lambda \right)\times a_{chla}^{*}\left(\lambda \right)]$ by immersing their instruments in a monospecies culture or a dye, constructing the curve [chla] ~ [Fchl] and assuming that this term is constant regardless of the conditions.

The dataset used by Roesler includes CTD, moorings, and profilers, but all the data during which the PAR had values greater than or equal to 200 µ Einstein m^−2^.s^−1^ (equivalent µmol m^−2^.s^−1^) are discarded to avoid quenching (this is the value from which the yield drops by more than 10%).

Roesler calculated the calibration coefficient of a sensor successively immersed in different cultures of mono-species algae. As shown in Figure 2, the calibration slope of optical sensors varies according to the species of algae encountered. He also showed that dissolved organic matter (CDOM) contaminates the signal at usual chlorophyll wavelengths. This results in an overestimation of in situ chlorophyll.

Roesler conducted a literature review to find the various calibration slopes mentioned by other authors. He listed them and noticed that there are relatively constant coefficients (slope factor) by geographical area, as shown in Table 1.

Roesler compiled the values to obtain an average by ocean basin, as shown in Figure 3.

Roesler established that the slope factor varies more according to the species than according to the physiological state for the same species, while the Wetlabs calibration coefficient is calculated for a single species in a single type of growing condition.

Finally, he recommends, without sampling possibilities, to use a factor of 2 to calibrate the Wetlabs.

Regarding laboratory in situ sample analysis, several techniques exist between trichromatic spectrophotometric methods, monochromatic with acidification, fluorimetry, and high-performance liquid chromatography [23,24,25,26].

Jeffrey et al. [27] present the results of an experiment carried out within the framework of the 78-SCOR working group, the aim of which was to compare the performance of these methods. The samples used were pure pigment solutions, algae cultures, and natural samples. They conclude that the trichromatic method in spectrophotometry allows for the accurate quantification of chlorophyll *a* in samples free of degradation products.

Methods with acidification, such as the monochromatic spectrophotometric method and fluorimetry, reduce but do not eliminate pheopigment-induced interferences. Neither of the two previous methods can distinguish chlorophylls from their precursors. The only method allowing an accurate quantification of chlorophylls in the presence of their precursors and degradation products is the separation chromatography technique (HPLC). Therefore, we agree with the choice of HPLC as the reference method for the analysis of chlorophyll *a* in seawater samples.

All these studies only partially address our problem. In particular, we have not observed, in our sea campaigns, either the influence of CDOM or the perturbation due to non-photochemical quenching, so we do not make any particular corrections on the raw values of our sensors. On the other hand, we keep the practice of checking the linearity of the sensor before deployment and calculating the slope factor after deployment by comparison with seawater samples.

## 3. Materials and Methods

The data presented in this work were collected during campaigns carried out between 2012 and 2020.

The equipment consisted of an SBE911 + CTD (Figure 4), a reference SBE35 probe to check the validity of temperature measurements, and a carousel of 12 sampling bottles. Before deployment, the CTDs were calibrated in pressure, temperature, and conductivity in a metrology laboratory.

The values obtained in situ were compiled into .btl files generated by SBE data processing 7.23.2 (Seabird Company software), which is the data conversion and processing software for the Seabird SBE 9 CTD profilers. The software can be downloaded for free from the manufacturer’s website (https://www.seabird.com/software (accessed on 22 November 2022). The .btl files provide very accurate values, averaged around the trigger of the sample bottle closure.

Several types of instruments allowing the measurement of chlorophyll in vivo were used during these campaigns and some of them were lent to us by external laboratories. This is the case of the Fluoroprobe BBE, the Phyto-PAM, and the flow cytometer.

The Aquatracka III (Chelsea) is a sensor whose range is from 0.01 to 100 µg/L. There is a version with a maximum immersion of 600 m, and another at 6000 m. The excitation wavelength was set at 430 nm, and the emission wavelength at 685 nm.

The optics were fitted with a xenon lamp emitting flashes at 5.5 Hz (Figure 5). The signal was logarithmically amplified.

For this instrument, we used calibration equipment comprising two quartz cells 8 cm long, 6 cm wide and 5 cm deep. This device allows the calibration of the sensor in pure chlorophyll solutions according to the instrument manual and the Seabird manufacturer application note No. 39 (https://www.seabird.com/cms-portals/seabird_com/cms/documents/application-notes/appnote39Feb10.pdf (accessed on 18 December 2022)).

The WetStar (Wetlabs) works in flow. Its measuring range was 0.7 to 75 µg/L. Its maximum immersion was 600 m, the excitation wavelength was set at 460 nm, and the emission wavelength at 695 nm (Figure 6).

The ECO-FL (Wetlabs) has an adjustable range: 0.5 to 30–65–125 µg/L. Its maximum immersion was 6000 m, the excitation wavelength was set at 470 nm, and the emission wavelength at 695 nm (Figure 7).

Regarding calibration, the manufacturer specifies in application notes No. 41 and 62, available on the Seabird website, that the factory coefficients can be used to obtain approximate values. On the other hand, the calculation of the correct calibration coefficient must be carried out by comparison with the in situ values determined by an independent method in the laboratory, with this coefficient becoming not applicable if the planktonic population changes.

The FluoroProbe probe (BBE-Moldaenke) is a multi-spectral fluorimeter that allows in situ and real-time detection in the water column of various groups of phytoplankton, including cyanobacteria. The result of the measurement is expressed in µg of chlorophyll *a* per liter of water. Individual profiles during measurements were obtained for green, blue–green/cyanobacteria, diatoms/dinoflagellates, and cryptophytes. Possible interferences due to yellow substances were eliminated by a CDOM correction factor. The principle of measurement was based on the fluorescence emission of the pigments of the collecting antenna of photosystem II in successive stages, such as sequential excitation of the pigments at 6 wavelengths (370, 470, 525, 590, and 610 nm), measurement of the fluorescence emission of chlorophyll *a* at 680 nm, and obtaining a global fluorescence excitation spectrum. Then, a comparison was performed with reference spectra from calibration with different populations of characteristic algae. Finally, chlorophyll *a* was quantified by an algorithmic deconvolution and the result was assigned to different phytoplankton groups. The instrument can be rigged on a CTD profiler to perform profiles or be used on a laboratory bench. The critical point here is the calibration, which must be carried out by immersing the instrument in a perfectly controlled mixture of algae cultures characteristic of the four phytoplankton groups.

The Phyto-PAM (manufacturer Walz) is a bench-top instrument that collects a fluorescence excitation spectrum by simultaneous application of several wavelengths. An irradiance measurement is possible at the same time. A deconvolution of the spectra obtained gives values of photosynthetic yield by group of algae, values of chlorophyll *a* by groups, and an estimated value of total chlorophyll *a* for the mixture of species. In practice [31], the living sample was first placed in the dark in order to minimize its fluorescence yield with respect to photochemical conversion. Under the action of a very low light intensity, this minimum fluorescence yield, *F*_min_, was measured. Then, a flash of saturated light was emitted so as to minimize the chemical conversion (by closing the reaction centers of photosynthesis) and to maximize the fluorescence. The *F*_max_ was then measured. Fluorescence measurements used the PAM (pulse amplitude modulated) method. The sensitive point is the calibration, which should preferably be performed with algal assemblages closest to those encountered in the natural environment.

The flow cytometer (Figure 8) is an instrument that makes it possible to move particles or cells at high speed in the beam of a laser, to count them, and to characterize them. It is the re-emitted light (by diffusion or fluorescence) which makes it possible to classify the population according to several criteria and to sort them.

The principle involves analyzing the optical or physical signals emitted by a particle cutting the light beam of a laser or an arc lamp. The signals measured are related to the intrinsic optical properties of the particles: they correspond to light-scattering phenomena linked to the dimensions of the particle, to their internal structure, or to the fluorescence of phytoplankton cells. These signals separated by optical filters were collected by photomultipliers (PMT), amplified, digitized, processed, and stored by a computer through a dichroic mirror and an optical filter. This individual analysis process (cell-by-cell) is multiparametric and can be performed at the speed of several thousand events per second. The computer calculated the statistical data associated with the distributions of the measured parameters and represented them in the form of histograms (one parameter) or cytograms (two parameters) on one or more populations, whose cellular properties were thus evaluated. The sorting function of the most advanced flow cytometers makes it possible to physically sort one or two cell populations defined by their optical properties. Flow cytometry makes it possible to count the different populations of photosynthetic phytoplankton on the basis of the fluorescence of phytoplankton pigments.

In addition to in situ measuring devices, seawater samples were taken according to usual practices as 1 to 2 L of water filtered through 25 mm GF/F.

Usually, 3 to 5 samples were taken in the photic zone, i.e., between 200 m and the surface, and 1 sample was taken below 200 m, at a depth where we were sure not to find algae. This last point constitutes the zero point.

The filters were kept in liquid nitrogen until arrival at the laboratory and then stored at −80 °C until extraction.

The technique used for measuring chlorophyll *a* was high-performance liquid chromatography (HPLC) using a Dionex Ultimate3000, with the method of Laurie Van Heukelem et al. [33]. The extraction of the pigments was carried out in 95% acetone by ultra-sonication for 10 s, followed by a phase of 12 h at −20 °C. The addition of an internal vitamin E tracer made it possible to calculate the extraction yield. The extract was injected onto a reverse-phase C8 column. Elution was carried out by a solvent gradient (mixture of 70% methanol and 30% tetrabutyl ammonium acetate) over a period of 31 min. Compounds were detected by DAD at 430 and 667 nm. The internal tracer was detected at 222 nm. The specificity of the method, in quantifying phytoplankton pigments, was demonstrated by international inter-laboratory [34,35,36,37,38] and a national exercise in 2014 between five French laboratories.

These round-robins have established criteria for estimating the quality of phytoplankton pigment analyses by HPLC. These criteria include precision (Prec) and accuracy (Acc) of the pigment measurement, reproducibility of retention times, resolution between zeaxanthin and lutein, and the quality of chlorophyll *a* calibration.

These parameters, summarized in Table 2, are quality indicators for measuring pigments. From the “Routine” category to the “State-of-the-Art” category, the results are increasingly accurate, sensitive, and precise. For in situ calibrations, it is recommended to be in the “State-of-the-Art” category, which is the case for our laboratory with a score of 3.5.

We conducted experiments, grouped in Table 3. 

## 4. Results

### 4.1. PROTEVS Cruises, September 2012 and 2013, Bay of Biscay (Figure 9)

The sensors rigged on the CTD were a WetStar and an Aquatracka III. The sensors were calibrated by immersion in increasing concentrations of mono-species algae cultures (*Dunaliella* species in 2012, *Tetraselmis suecica* in 2013). For each culture concentration, one liter of solution was taken and filtered through 25 mm-diameter glass fiber filters, type GF/F.

**Figure 9 sensors-23-02825-f009:**
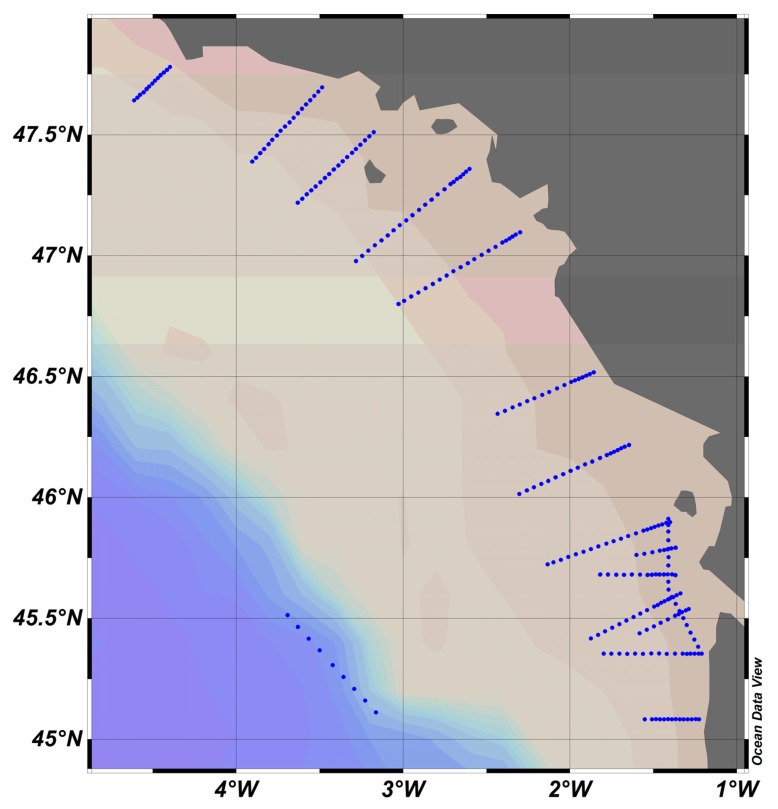
PROTEVS stations map, where blue dots represent stations. French coast on the right, close to the town of Nantes.

An HPLC analysis was performed on the sample and the chlorophyll *a* sensor ~ chlorophyll *a* HPLC curve is plotted in Figure 10.

The implementation of algal cultures for sensor calibration presents many problems. First, our laboratory does not have a facility that allows the culture. We were obliged to get our supplies from outside and to work as fast as possible in order not to degrade the culture. In addition, the volume of secondary standards was too large to be able to multiply the calibration levels. This is why our 2012 and 2013 calibrations only have three levels. Even so, the impact on our calibration was limited because a small number of trials affects the uncertainty more than the slope of the regression.

In Figure 9, it is observed that for the same instrument, the slopes differed from one culture to another. It was not possible to establish a robust calibration coefficient. In situ calibration also revealed large variations, thus confirming the literature data (Ricour et al. [39]).

When the pre-campaign calibration was completed, the sensors were deployed at sea with their new slope factor. One liter of seawater was taken from the Niskin bottles and the samples were filtered through GF/F filters, and the filters were kept in liquid nitrogen. Upon return to shore, the samples were analyzed by HPLC.

If the calibration performed in the laboratory prior to the cruise was suitable, a slope approaching 1 should be observed between the sensors and the HPLC analyses. This was not the case, as shown in Figure 11.

The correlation coefficients ranged from 0.75 to 0.86, demonstrating a linear relationship between the sensor values and the laboratory analyses.

However, when we look at the slopes of the calibration, they are significantly different from 1, which means that the pre-calibration was not suitable and that we needed to repeat the calibration using the samples collected in situ. The slope factors were not constant between sample dates (cruises) and instruments.

We plotted the slope factor ~ time, i.e., the drift of the instrument over time. In the ideal case, this drift is linear. However, concerning our sensors, this was not the case, as shown in Figure 11. These variations were not due to a possible drift of the instruments because in this case, the slopes would decrease or increase linearly with time. What we observed in Figure 12 is a random error due to the calibration conditions. However, the only parameter that has been modified here is the reference algal composition. It can be deduced that the pre-campaign calibration in algal cultures was not suitable.

A particular shift was observed in May 2013 when the sensors were calibrated in the laboratory. As the slope factor is defined by the relationship between the chla sensor and the chla HPLC, it can be seen that the fluorescence yield was lower for both sensors (1/3 for the WetStar and 1/7 for the Aquatracka III). The only operating condition that was modified was the algal strain (taxon *Tetraselmis*), with an uncertainty on the good physiological state of the cells because we do not have suitable facilities in the laboratory.

### 4.2. 2016 DYNSEDIM Campaign, March, Southern Brittany

This campaign was used to test several types of biomass measurement instruments.

In addition to the Aquatracka and WetStar fluorometers, we used a Fluoroprobe BBE probe from Moldaenke and a Phyto-PAM phytoplankton analysis system from Walz.

We paralleled three high-frequency in situ measurement instruments (Aquatracka, WetStar, Fluoroprobe), an in vivo punctual measurement instrument (Phyto-PAM), and the measurement of chlorophyll *a* by HPLC.

Figure 13 shows the comparison between the different instruments at stations 5 to 9. The instruments installed on the CTD profiler were plotted as lines and represent continuous measurements. We took three samples from each profile, at the surface, at the maximum chlorophyll concentration, and at the bottom. We then analyzed the samples both by HPLC and using Phyto-PAM. These discrete measurements are represented by dots, black for HPLC and red for Phyto-PAM. Consistency was observed between the Aquatracka and WetStar sensors. There was a correlation between the HPLC and Phyto-PAM analyses. The Fluoroprobe was not consistent with either the Aquatracka or WetStar sensors or with the laboratory analyses. Later in the campaign, its malfunction increased, probably due to a pressure problem.

Figure 14 represents the chlorophyll *a* value of each instrument versus the chlorophyll *a* value determined by HPLC.

The slope factor showed an excellent correlation between Phyto-PAM and HPLC analyses (coefficient r² above 0.9). This result agrees with the Phyto-PAM measurement principle, which considers the performance of different classes of algae with respect to photosynthesis. Nevertheless, the Phyto-PAM overestimated the real chlorophyll content, which seems to indicate a calibration error of the instrument. There was also a strong correlation between the Aquatracka III, WetStar, and HPLC analyses (r² > 0.8). At the end of this campaign, it appears that we have improved since the tests conducted in 2012–2013, notably by optimizing our method of analysis of chlorophyll *a* by HPLC. We adapted the method of Van Heukelem et al. to our analytical system and optimized the technique by looking for the best compromise between the eluent flow rate and column temperature to obtain good resolution and higher sensitivity.

### 4.3. 2017 RESOMAR–COAST-HF Technical Workshop Dedicated to the Measurement of Fluorescence

This was an inter-comparison exercise between sensors on a buoy called MOLIT located in the bay of Vilaine (Brittany, France). Reference chlorophyll analyses were performed using HLPC and a benchtop fluorimeter.

The instruments used during the inter-comparison were all based on optical measurements, some scanning several wavelengths, such as a BBE Fluoroprobe. Two MP6-NKE multi-parameter probes, which measure subsurface temperature, salinity, oxygen saturation, turbidity, and fluorescence, and a flECO-AFL from Wetlabs, were also used.

At the same time, the samples were taken in duplicate and finally analyzed with a laboratory fluorimeter and HPLC. Results obtained by all these instruments are displayed in Figure 15.

From this figure, a relative coherence between the different sensors can be observed, but the correlation matrix calculated between all the techniques shows the difficulty of establishing a linear relationship (see Figure 15).

As shown in Table 4, the in situ measurements were poorly correlated with each other. Chlorophyll measurements by fluorimetry were insufficiently correlated with chlorophyll measurements by HPLC.

Laboratory assays were poorly correlated with sensor values, apart from the case of flECO-AFL.

### 4.4. MOCOSED 2017 Cruise, North Atlantic Zone, Greenland Sea, Norwegian Sea in September–October (Figure 16)

Before deployment at sea, we carried out a metrological verification of our instruments by measuring the minimum voltage obtained by closing the optical windows of the instrument with black tape (dark count), then checking that the maximum voltage was close to 5 volts by applying a solid fluorophore to the optical windows. Finally, its linearity was checked by measuring an increasing range of rhodamine solutions, as shown in Figure 17.

**Figure 16 sensors-23-02825-f016:**
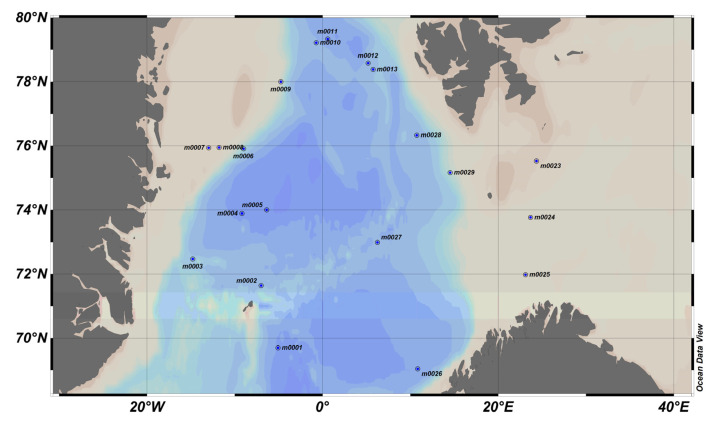
MOCOSED 2017 area, where blue dots represent stations. On the left, east coast of Greenland, bottom right, north of Norway, and at the top, south coast of Svalbard.

When the sensor was launched, the calibration coefficients were those of the manufacturer and the profiles were only corrected at the end of the campaign, by analyzing the samples taken using Niskin bottles.

When plotting the chla sensor ~ chla HPLC regression line, we obtained a slope factor of 5.6573 and a correlation coefficient of 0.83, as seen in Figure 18.

We can assume that by calculating the calibration slopes profile by profile, we were free from a large part of these biases. The principle of relative spatio-temporal constancy of the species and the environment along the water column was established.

Based on this principle, we increased the number of sampling points along the profile to have 3 to 6 points from the lower limit of the photic zone (in principle, no chlorophyll *a*) to the surface.

We then calculated the slope factor term = chlorophyll sensor/chlorophyll lab, which allowed us to correct the in situ profiles (see Figure 19).

It can be observed that at all the stations, the in situ values and the laboratory values were only correlated with an R^2^ of 0.83, whereas by calculating the slope factor, calculated station by station (see Figure 20), it allowed a linear relationship, with R^2^ varying from 0.92 to 1.

The slope factor values varied from 9.45 to 2.93. This result is consistent with the observations made so far. Since the fluorescence yield of the instruments is closely dependent on the variability of the medium, it did not seem relevant to correct a profile carried out at the end of August in the Greenland Sea and a profile carried out in October near the coast of Norway with the same coefficient.

### 4.5. Gibraltar 2020 Campaign, Western Mediterranean, October (Figure 20)

During the campaign, samples were taken for chlorophyll analysis at the rate of 6 levels per profile in the area between approximately 200 m and the surface.

Calculating a single slope factor for all profiles yielded a mean of 2.08 and an R^2^ of 0.96. This result is already very good because two thirds of the samples came from a very homogeneous body of water east of the Strait of Gibraltar. The uncertainty on the slope factor, calculated according to Student’s law and by the excel formula LOI.STUDENT. INVERSE.BILATERAL, yielded a value of 0.2 (Figure 21).

The linear model was very good, but the calculation of the slope factor profile by profile (Figure 22) made it possible to increase this R^2^ to an average of 0.98, and especially, we note that the stations did not have the same characteristics with regard to the measurement of the fluorescence.

The comparison of the slope factors obtained from the literature [13] yielded a value of 1.6 ± 0.3 in the Ligurian Sea and 1.7 ± 0.2 in the Cretan Sea. These values are similar to those obtained on the transect located east of the Strait of Gibraltar, as shown in Figure 23.

The slope factor can be used to distinguish between different water bodies, better than the raw sensor values. In particular, Figure 23 shows that in the Atlantic zone, the slope factor confirmed the influence of the nutrient stock on the fluorescence yield.

Indeed, it is known that the availability of nutrients has an impact on the composition of phytoplankton communities and their physiological state. For example, diatoms, which are fairly well-represented in this area, have a great need for silicates to build their silica envelope. The stock is rapidly depleted during the spring and autumn blooms in the Mediterranean, east of the Strait of Gibraltar. In contrast, the stock is renewed in the west by the Atlantic Ocean. In the east, the population is weaker and in a worse physiological state than in the west.

This will result in a different fluorescence yield depending on the area. This variation in fluorescence yield is not visible west of the Strait of Gibraltar if we only consider the sensor values. On the other hand, the slope factor perfectly reflects the characteristics and composition of the environment, as can be seen in Figure 24.

## 5. Discussion

We have seen the complexity of expressing in situ fluorescence measurements in chlorophyll concentration. However, the experiments we conducted made it possible to better understand and characterize the instruments used. They also made it possible to develop a metrological monitoring protocol for sensors as well as a robust in situ calibration procedure.

This protocol includes the measurement of the dark count by sealing the optical windows of the instrument with black tape, the measurement of the span of the range at approximately 5 volts, and the verification of linearity by a range of rhodamine. This was performed before the campaign. At this point, the sensor was integrated into the CTD profiler with the current dark count value measured in the laboratory and the manufacturer’s slope value.

During this phase, it is also important to define user tolerances beyond which the instrument must be returned to the manufacturer for repair.

During the campaign, an approximation of the chlorophyll *a* content can be obtained by arbitrarily dividing the Wetlabs sensor values by 2, as recommended by Roesler et al. [13].

We observed a weak impact of our instrumentation with respect to non-photochemical quenching. This effect was probably due to the position and orientation of the sensors under the Niskin bottle rosette, as well as the flow operation of the WS3S sensors. It would be relevant to carry out sea trials by testing various configurations in order to precisely quantify this aspect. It could be seen that, depending on the geographical areas and the campaign period, non-photochemical quenching affects the chlorophyll measurements. To this end, we recommend the systematic use of a quantimeter in parallel with the measurements of chlorophyll and CDOM in order to be able to correct the profiles a posteriori.

Considering the mastery acquired in the analysis of pigments in HPLC, we confirmed this technological choice for the reference measurement. In this case, the samples were analyzed upon return from the campaign, provided that there is either a reserve of liquid nitrogen or a −80 °C freezer on board to ensure their conservation.

By following this procedure: verification of the dark count, linearity, max output before deployment, then sampling with the Niskin bottle, conservation of the filters in liquid nitrogen, HPLC measurements, and comparison with the .btl files from the CTD, it was possible to obtain a robust and precise in situ calibration with an uncertainty of 0.2–0.3 on the slope factor and a correlation coefficient greater than 0.95.

Finally, we note that the slope factor value made it possible to better characterize the water masses than the in situ fluorescence values as recently reported by Petit et al. [40].

In conclusion, we can say that pre-campaign calibration in algae culture is not recommended. Even with a mixture of algae as close as possible to the one encountered in the field, one cannot predict the physiological state of the cells or the constituents of the seawater that will interfere with the measurement. Only an in situ calibration carried out by comparison with the HPLC analysis of samples taken from the same area allows true values to be obtained.

## Figures and Tables

**Figure 1 sensors-23-02825-f001:**
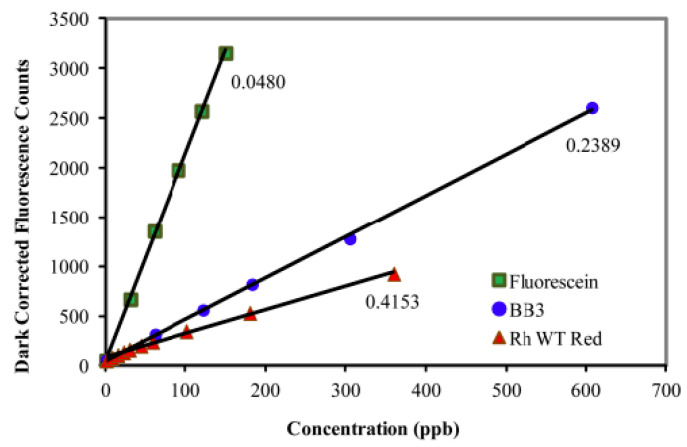
Fluorescence (λ_exc_ = 470 nm, λ_em_ = 695 nm) of the ECOTriplet Chl-a sensor (Wetlabs) at increasing concentrations of fluorescein, Basic Blue 3, and rhodamine WT red. Reprinted with permission from [7] The Optical Society.

**Figure 2 sensors-23-02825-f002:**
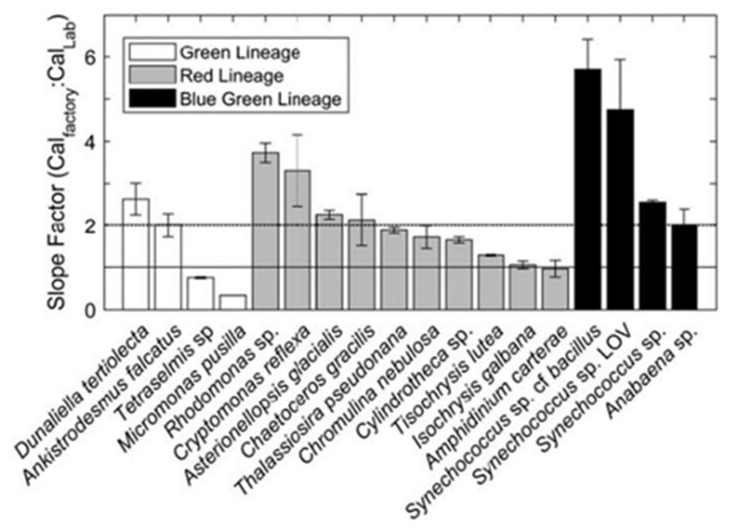
Factory coefficient/in situ coefficient ratio according to different types of algae. Reprinted with permission from Roesler 2017 [13] CC BY 4.0).

**Figure 3 sensors-23-02825-f003:**
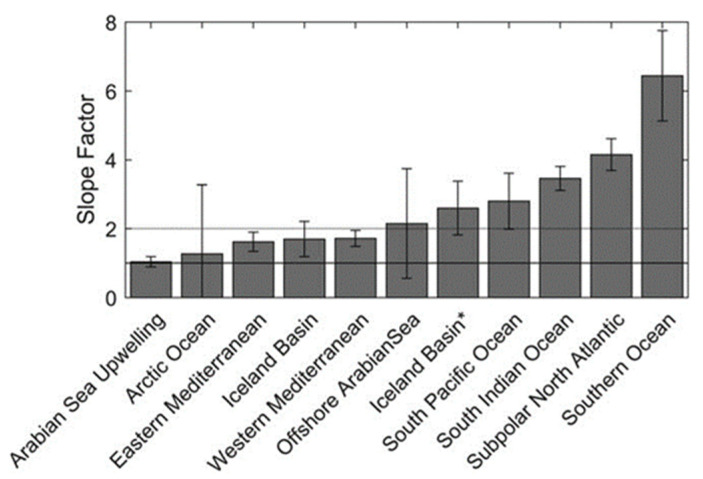
Average values of the slope factors by geographical area. Reprinted with permission from Roesler 2017 [13] CC BY 4.0. * Two bars are visible for the Icelandic basin. The first one is obtained by using FLNTU and ECOBBFL2 sensors (slope factor 1.6), the second one by using an ECOTriplet sensor (slope factor 2.6) as explained in Table 1.

**Figure 4 sensors-23-02825-f004:**
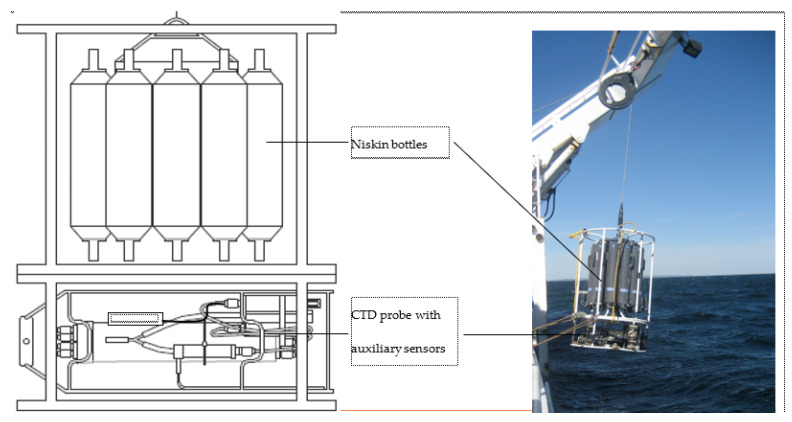
CTD probe and sensors with sampling Niskin bottles.

**Figure 5 sensors-23-02825-f005:**
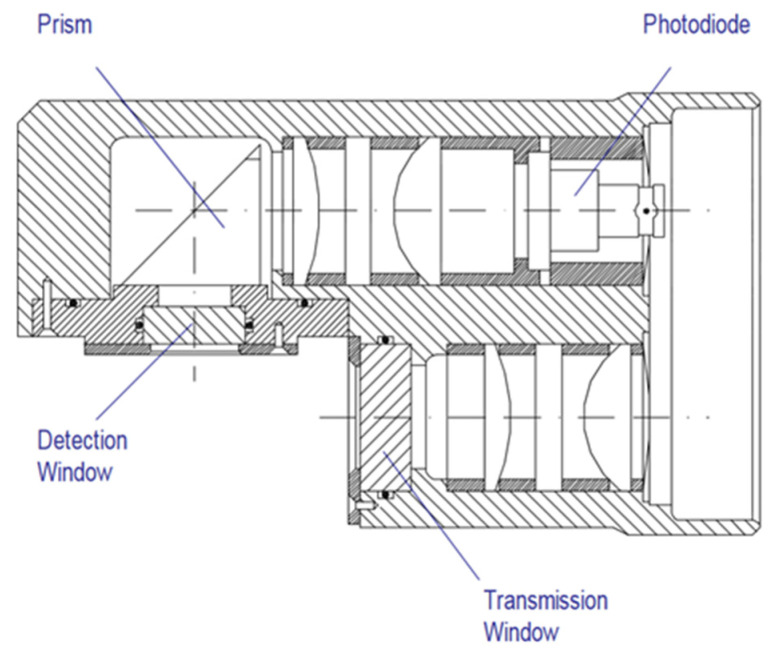
Aquatracka III optical path. Reprinted with permission from [28] Chelsea Technologies Group.

**Figure 6 sensors-23-02825-f006:**
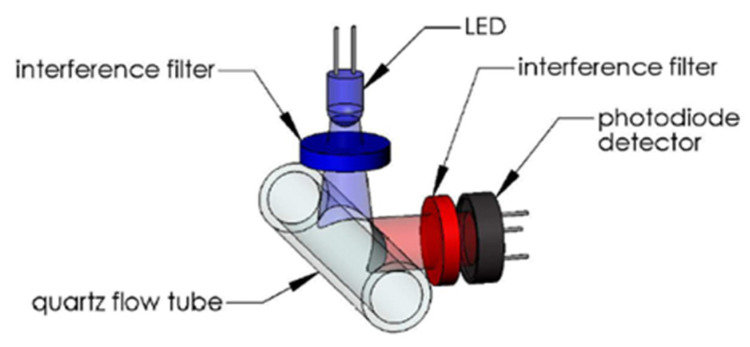
WetStar optical path. Reprinted with permission from [29] Sea-Bird Electronics.

**Figure 7 sensors-23-02825-f007:**
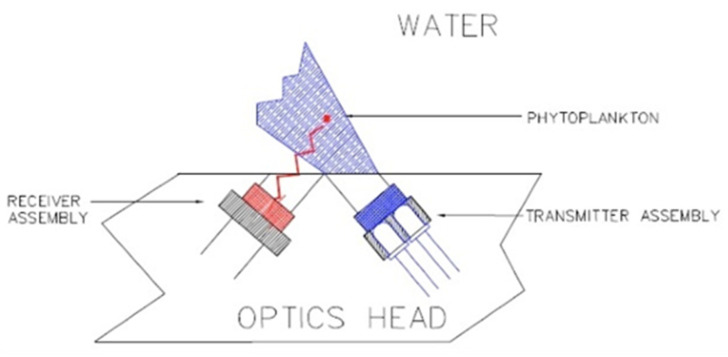
ECO-FL optical path. Reprinted with permission from [30] Sea-Bird Electronics.

**Figure 8 sensors-23-02825-f008:**
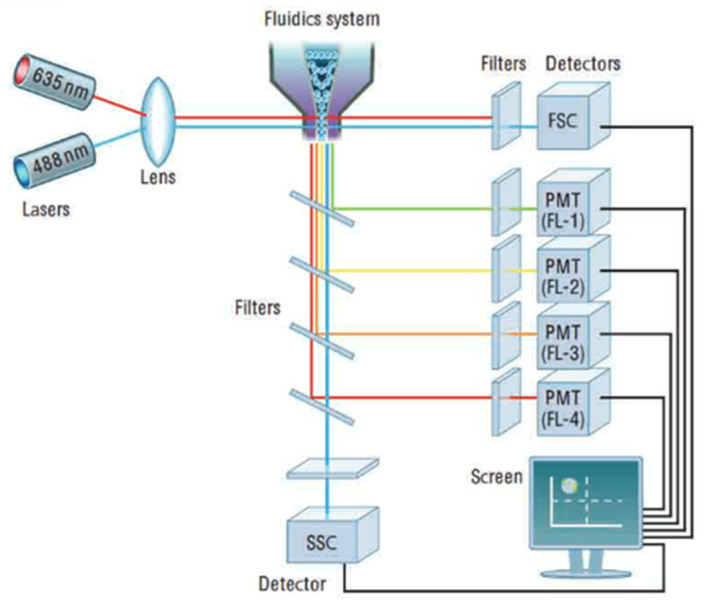
Cytometer optical device. Reprinted with permission from [32] AbD Serotec.

**Figure 10 sensors-23-02825-f010:**
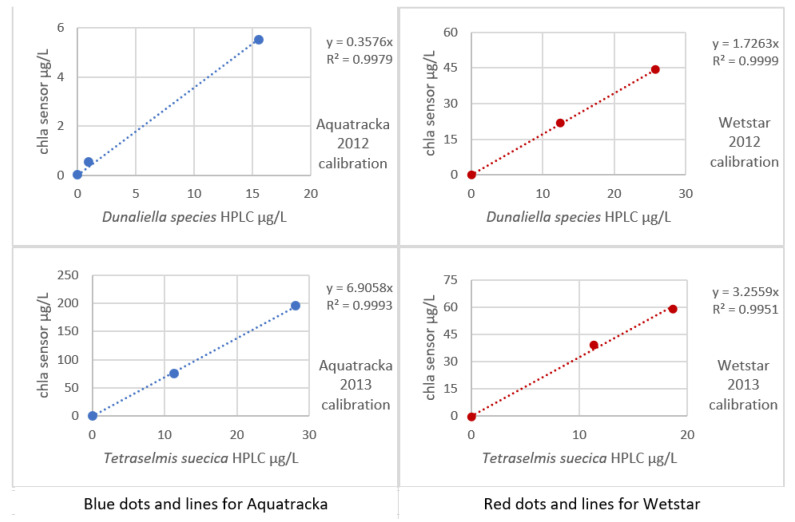
Sensor calibration curves (2012) in mono-species *Dunaliella* and (2013) in mono-species *Tetraselmis*.

**Figure 11 sensors-23-02825-f011:**
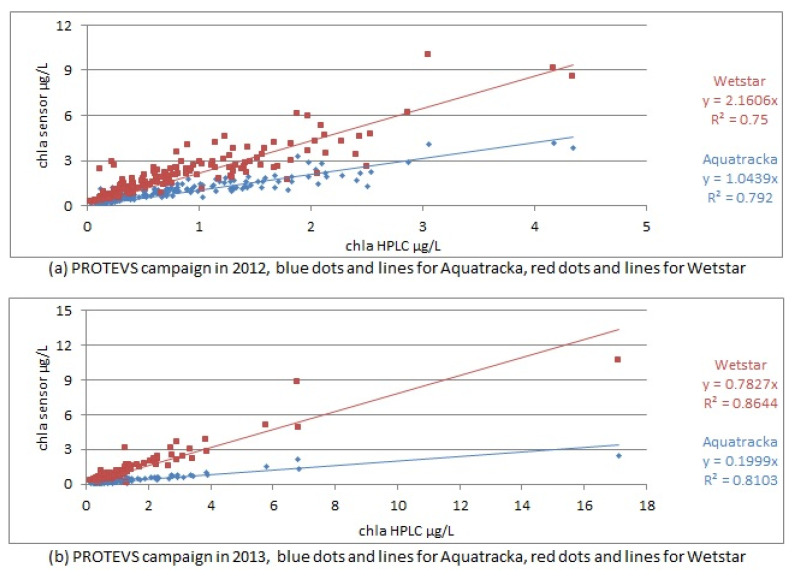
In situ calibration curves for PROTEVS cruises, Image above in 2012, below in 2013.

**Figure 12 sensors-23-02825-f012:**
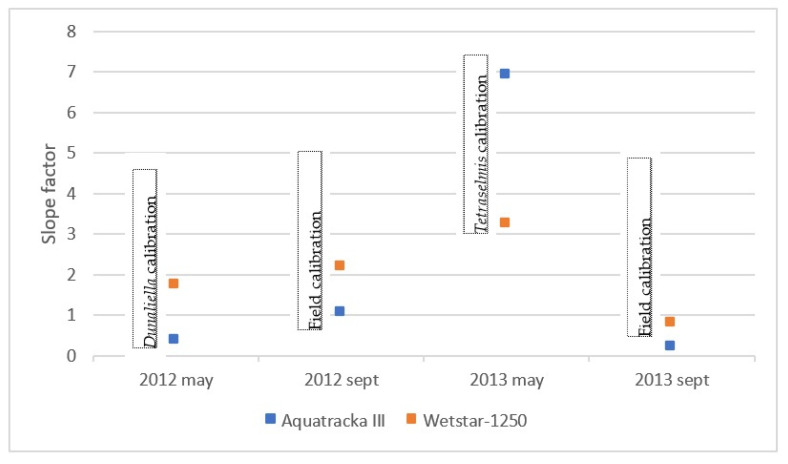
Report of the slope factor observed for each sensor over two years.

**Figure 13 sensors-23-02825-f013:**
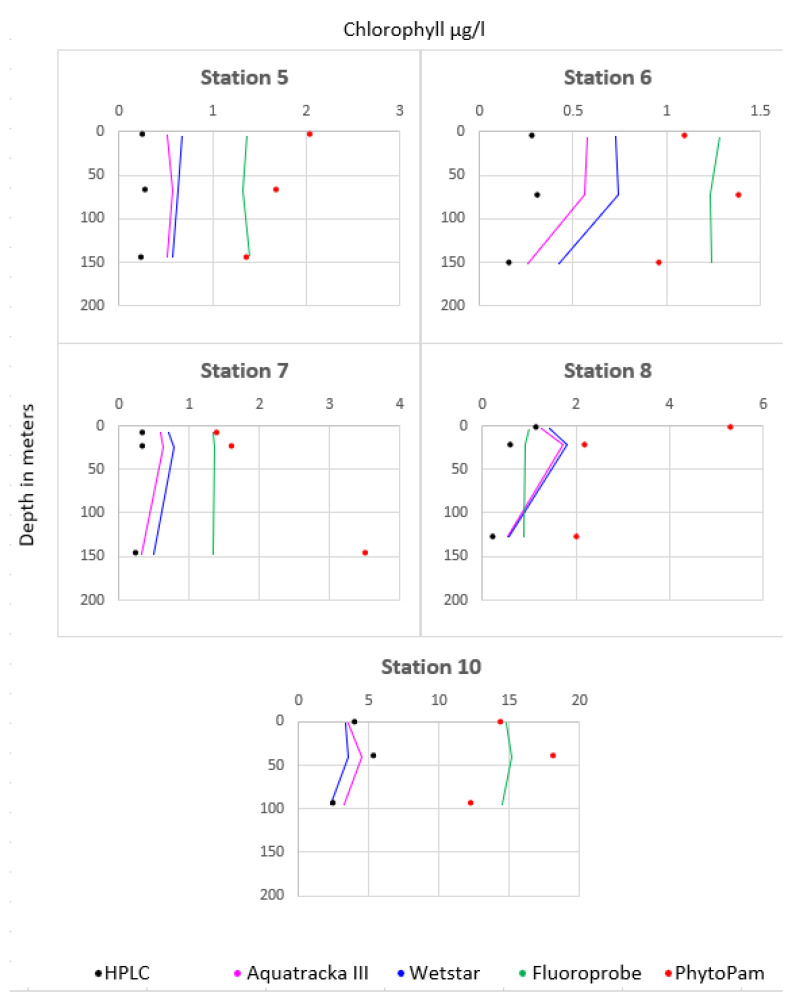
Values recorded by all the sensors during the DYSEDIM 2016 campaign.

**Figure 14 sensors-23-02825-f014:**
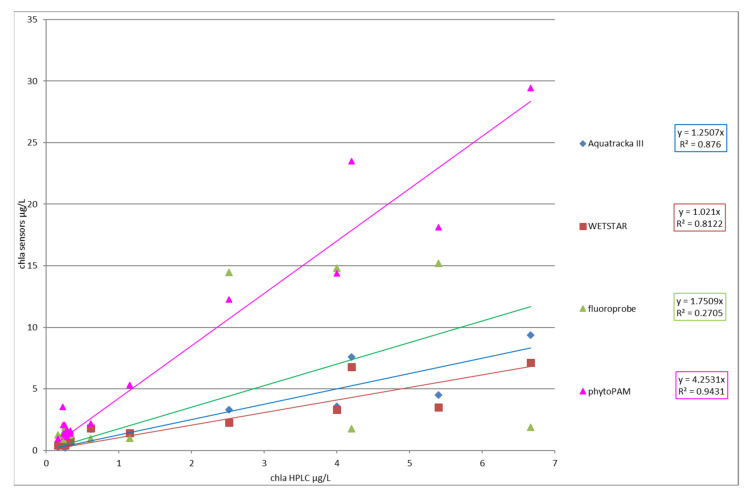
Slope factor (chla_sensor_/chla_HPLC_) for each instrument.

**Figure 15 sensors-23-02825-f015:**
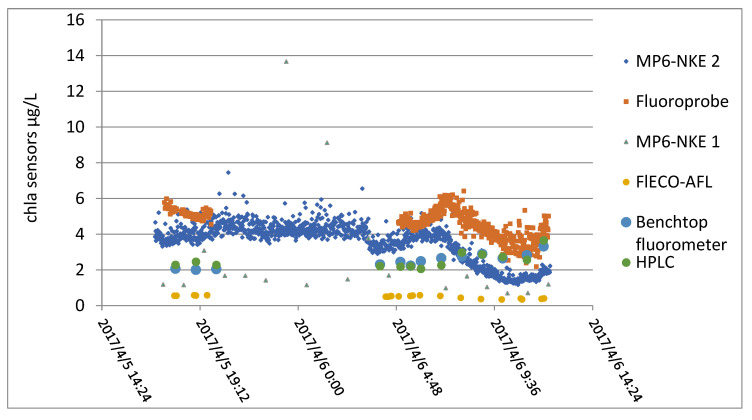
Values obtained with all the sensors.

**Figure 17 sensors-23-02825-f017:**
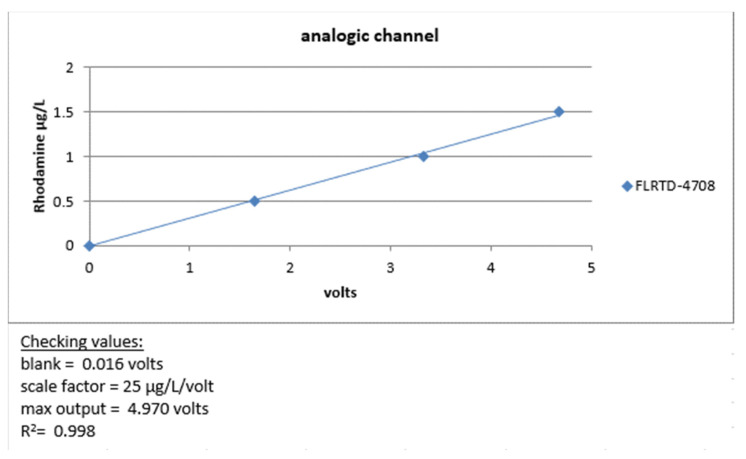
FLRTD-4708 calibration report 2017.

**Figure 18 sensors-23-02825-f018:**
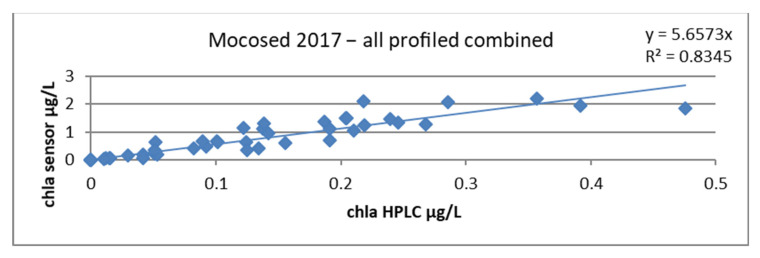
Relationship between chlorophyll concentrations obtained by HPLC and by the sensor, all stations combined during the MOCOSED 2017 campaign.

**Figure 19 sensors-23-02825-f019:**
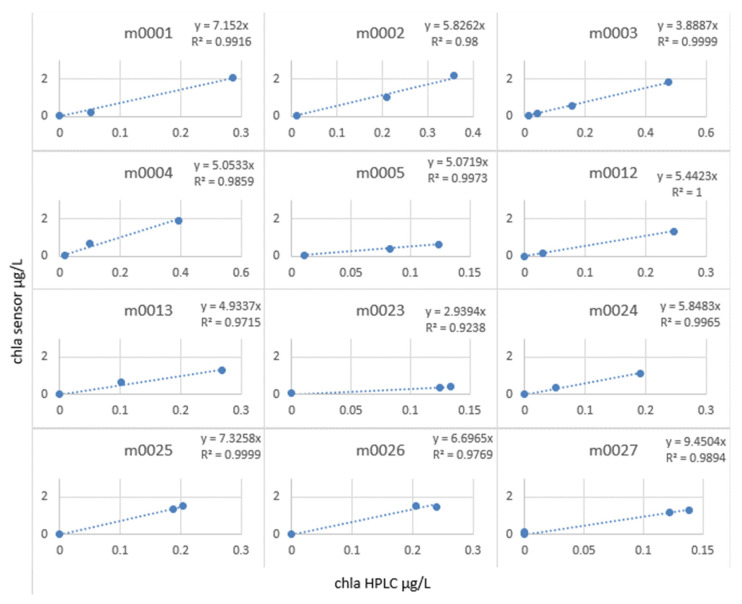
Slope factor by station calculated for MOCOSED 2017 campaign.

**Figure 20 sensors-23-02825-f020:**
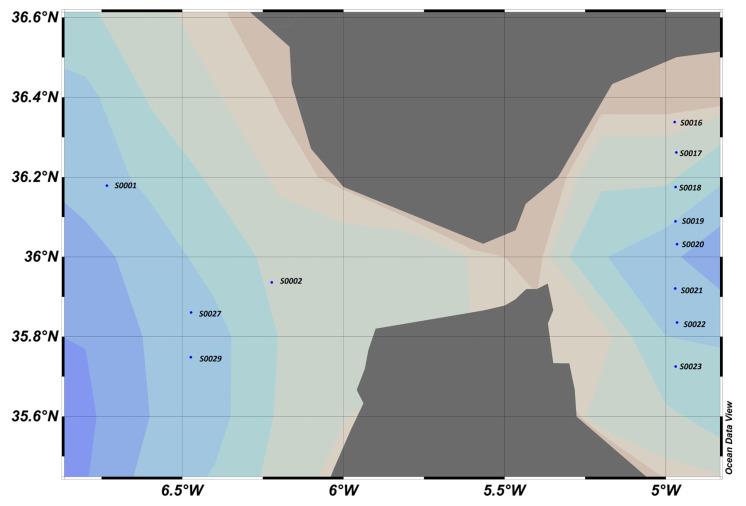
Gibraltar 2020 area, where blue dots represent sampling stations.

**Figure 21 sensors-23-02825-f021:**
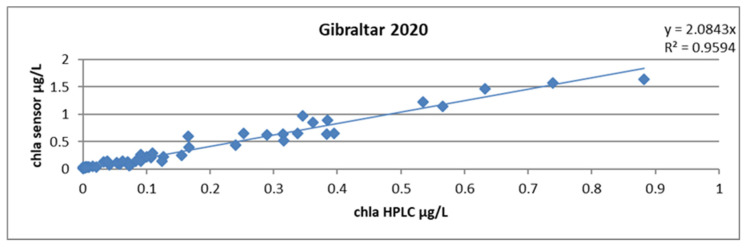
Relationship between chlorophyll concentrations obtained by HPLC and by the sensor, all stations combined during the GIBRALTAR 2020 campaign.

**Figure 22 sensors-23-02825-f022:**
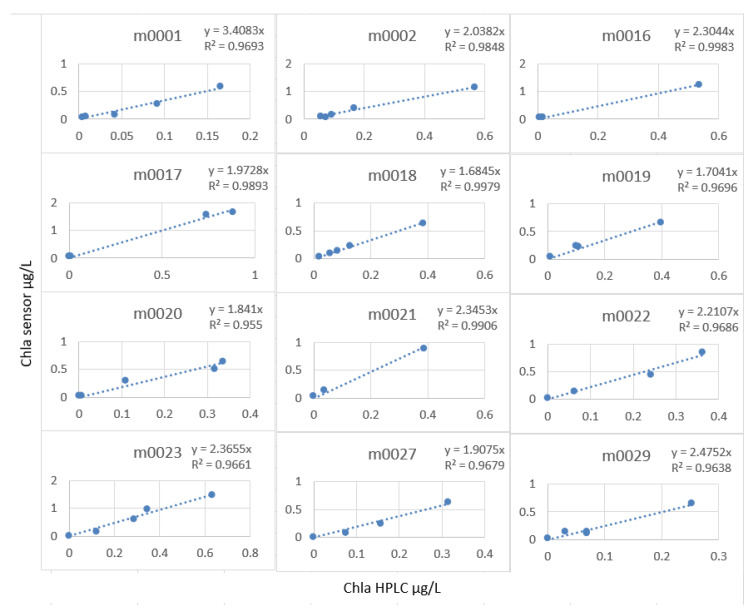
Slope factor by station.

**Figure 23 sensors-23-02825-f023:**
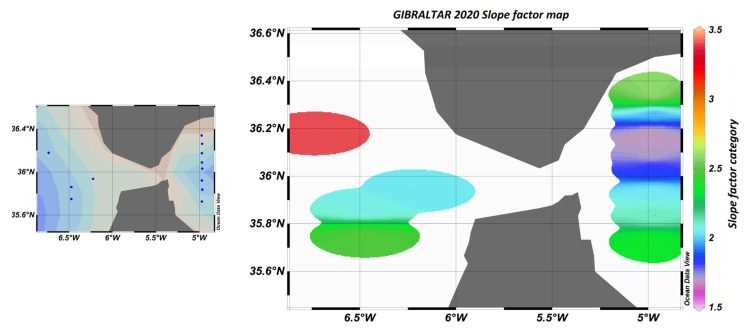
Map of sampling points and “slope factor” determined by calculation, where blue dots represent sampling stations.

**Figure 24 sensors-23-02825-f024:**
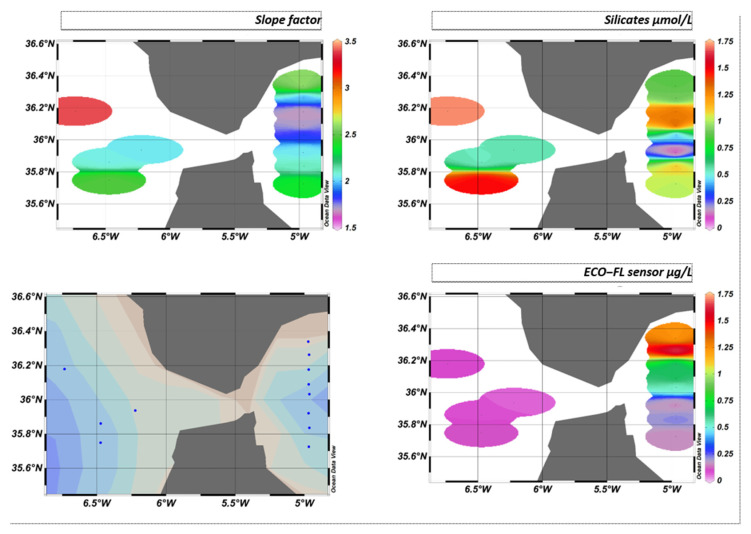
GIBRALTAR cruise iso-surfaces for slope factor, silicates, and ECO-FL sensor, where blue dots represent sampling stations.

**Table 1 sensors-23-02825-t001:** Review of the different regional slope factors obtained in the literature. Adapted with permission from Roesler 2017 [13] CC BY 4.0).

Project/Cruise	Oceanic Province	Lat/Lon Ranges		Dates	Slope Factor (HPLC)	Source(s)
MALINA	Arctic Ocean (27)	71 N	127 W	Jul 2007	1.27 ± 2.01	Coupel et al. (2015) [14]
NAB08	Iceland Basin (25)	60–62 N	25–28 W	May 2008	1.70 ± 0.51 2.60 ± 0.78 (*)	Cetinić et al. (2015) [15]
GEOVIDE NAT-LAS	Subarctic Atlantic Ocean (24)	56–60 N 58 N	27–39 W 51 W	Jun 2014 May 2013	4.15 ± 0.46	unpublished results
Bio-ArgoMED BOUSSOLE DEWEX MOOSE	Western Mediterranean (43)	41–44 N 43 N 42 N 41–42 N	7–12 E 8 E 5 E 5–6 E	May 2015 Jul 2013 Feb–Apr 2013 Jul 2014	1.62 ± 0.28	unpublished results Antoine et al.(2008) [16] Lavigne et al. (2015) [17] D’Ortenzio et al. (2014) [18] Lavigne et al. (2015) [17]
Bio-ArgoMED BOUM	Eastern Mediterranean (43)	34–38 N 34 N	19–29 E 33 E	May 2015 Jun 2008	1.72 ± 0.23	unpublished results Crombet et al. (2011) [19]
SS286	Arabian Sea offshore gyre (32) Monsoonal upwelling (22)	18–21 N 21–22 N	67–70 E 66–70 E	Mar 2011	2.15 ± 1.59 1.04 ± 0.15	Do Rosário Gomes et al. (2014) [20] Thibodeau et al. (2014) [21]
OUTPACE	South Pacific Ocean (51)	19 S	165–171 W	Mar 2015	2.80 ± 0.81	unpublished results
SOCLIM	South Indian Ocean (52)	43–53 S	52–72 E	Jan 2015	3.46 ± 0.35	unpublished results
SOCCOM	Southern Ocean (23, 53)	39–68 S	13 E–144 E	Mar 2014–Mar 2016	6.44 ± 1.31	Schuller et al. (2015) [22] Boss and Haëntjens (2016) [10]
	Global Data set mean Data set median Areally-averaged mean				2.63 ± 0.29 2.15 4.00 ± 0.48	

(*) Three models of fluorometers were used on the NAB08 cruise; FLNTU and ECOBBFL2 both exhibited a slope factor of 1.7, while the ECOTriplet exhibited a slope factor of 2.6.

**Table 2 sensors-23-02825-t002:** Performance metrics for pigment measurement in HPLC (from [38], SeaHarre 5 round-robin).

Performance Weight, Category, and Score		Total Chla		Total Pigment		Separation		Injection, Ɛ_inj_		Chla Calibration	
		Prec%	Acc%	Prec%	Acc%	Rs	Ɛ_Tr_%	Peri%	Chla%	Res%	Ɛ_etal_%
1. Routine	0.5	8	25	13	40	0.8	0.18	10	6	5	2.5
2. Semi-quantitative	1.5	5	15	8	25	1	0.11	6	4	3	1.5
3. Quantitative	2.5	3	10	5	15	1.2	0.07	4	2	2	0.9
4. State-of-the-Art	3.5	≤2	≤5	≤3	≤10	≥1.5	≤0.04	≤2	≤1	≤1	≤0.5

**Table 3 sensors-23-02825-t003:** Experiments and datasets for 2012–2020.

Year	Cruise	Location	Period	Instruments	Pre-Campaign Calibration	Sampling	Post-Campaign Calibration
2012	PROTEVS	Bay of Biscay	Sept	WetStar	*Dunaliella* sp.	2 samples per station (surface and maximum chlorophyll)	Sampling + HPLC (all stations mixed)
				Aquatracka			
2013	PROTEVS	Bay of Biscay	Sept	WetStar	*Tetraselmis suecica*	2 samples per station (surface and maximum chlorophyll)	Sampling + HPLC (all stations mixed)
				Aquatracka			
2016	DYNSEDIM	Southern Brittany	March	WetStar	None, using manufacturer coefficient	3 samples per station (surface, maximum chlorophyll, and bottom ~120 m)	Sampling + HPLC (all stations mixed)
				Aquatracka	None, using manufacturer coefficient		
				FluoroProbe *	Unknown		
				Phyto-PAM *	Unknown		
2017	RESOMAR	Bay of Vilaine	June	Eco-FL	None, using manufacturer coefficient	2 samples per station (surface and bottom ~15 m)	Sampling + HPLC + benchtop fluor (all stations mixed)
				MP6-NKE *	Unknown		
				FluoroProbe *	Unknown		
2017	MOCOSED	Greenland Sea, Norwegian Sea	Sept-Oct	Eco-FL	Metrological checking in rhodamine	3–4 samples per station (~100 m to the surface)	Sampling + HPLC (per profile)
2020	GIBRALTAR	Western Mediterranean	Oct	Eco-FL	Metrological checking in rhodamine	3–5 samples (200 m to the surface) + 1 sample below 200 m	Sampling + HPLC (per profile)

Instruments marked with * are equipment owned by the partners who participated in the field work.

**Table 4 sensors-23-02825-t004:** Correlation matrix of all techniques.

	Fluoroprobe	MP6-NKE 2	MP6-NKE 1	flECO-AFL	Benchtop fluorometer	HPLC
Fluoroprobe	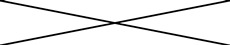					
MP6-NKE 1	0.006	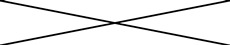				
MP6-NKE 2	0.105	0.116	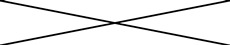			
flECO-AFL	0.326	0.670	0.215	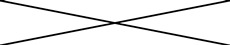		
Benchtop fluorometer	0.392	0.454	0.004	0.807	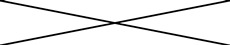	
HPLC	0.344	0.419	0.271	0.761	0.603	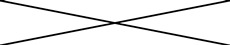

## Data Availability

The data presented in this study are available on request from the corresponding author. The data are not publicly available due to privacy.

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
