# Peer review of "In Situ Calibration of Wetlabs Chlorophyll Sensors: A Methodology Adapted to Profile Measurements"

_sensors, 2023, doi:10.3390/s23052825_

Round 1

Reviewer 1 Report

Reviewer Comments

The issue of in situ monitoring of chlorophyll a is an important environmental topic and this paper should make a strong contribution to the literature. In particular, the authors present quite a rich and interesting data set. However, before I can recommend publication, a number of issues must be addressed. I begin with some general observations and then proceed with specific comments/questions.

General Observations

The manuscript is readable, but the English requires polishing. I made suggestions in numerous areas directly on the manuscript file, but I could not decipher the meaning of a few passages, and these also have been marked (and in some cases, noted below).

In general, the manuscript reads more like a technical report than a scientific paper. Part of the issue is the formatting with numerous single sentence paragraphs. I suggested some examples to merge these single sentences into paragraphs at the beginning of the manuscript, but leave it to the authors to follow through on the remainder of the manuscript. The other part of the issue is the lack of detail in discussing some of the study results. More specific observations follow.

Specific Notes

1.       Suggest you add “in situ measurements” to the key words.

2.       Line 56, the term “industrial phase” is unclear.

3.       Line 81:  Since it is impossible to establish a robust calibration coefficient prior to field use, it 80 must be recognized that data collected in real time can only be qualitative in the absence 81 of concurrent sampling. Strictly, I would argue that the data are not qualitative, but are quantitative with some amount of bias, or perhaps quantitative with a higher degree of uncertainty.

4.       Lines 92-96 – These passages really are more Methods than a general Introduction.

5.       There really is not strong and clear paper objectives statement in the Introduction.

6.       Line 103 - However, feedback from users has 102 prompted manufacturers to now specify in most of their documents that only in-situ calibration with concomitant water sampling can approximate the actual chlorophyll value 104 in μg/l. I’m not sure this is statement is true for all manufacturers. Could you clarify and perhaps provide specific examples.

7.       Line 127 - … until the offset is caught up. I’m not clear on what you mean by this phrase.

8.       In conducting your “State-of-the-Art” assessment, how did you choose which papers you would reference? Did you use some form of semi-systematic review process that followed the Preferred Reporting Items for Systematic Reviews and Meta-Analyses (PRISMA) guidelines, for example (see also Selçuk, 2019; Snyder, 2019; Page et al., 2021)? The references tend to be a bit dated, with the most recent references being 2017.  Has there been no more recent work published?

Page, M. J., McKenzie, J. E., Bossuyt, P. M., Boutron, I., Hoffmann, T. C., Mulrow, C. D., ... & Moher, D. (2021). The PRISMA 2020 statement: an updated guideline for reporting systematic reviews. Bmj, 372.

Selçuk, A. A. (2019). A guide for systematic reviews: PRISMA. Turkish Archives of Otorhinolaryngology57(1), 57.

Snyder, H. (2019). Literature review as a research methodology: An overview and guidelines. Journal of business research104, 333-339.

9.       Line 179 – It is not clear what you mean by increases regularly. Monotonically?

10.   Equation 4 – how is FC defined?

11.   The transition to the fluorescence intensity formula on Line 201 is quite abrupt. Perhaps introduce the context of the equation a little better.

12.   Lines 214-216 – This sentence is long, convoluted, and difficult to follow. Please re-phrase. The same is true for Lines 254-261.

13.   Figure 2 actually should be Table 1.

14.   Methods – at what depths were the water samples from the carousel collected? Later, in the Results, it seems there may have been different depths, depending on the cruise effort. This should be noted in the Methods. Perhaps also in the Methods, you should expand information for the different cruises at different times and different locations and summarize this in a Table, to better set up your Results section.  This would help round out your final three paragraphs of the section.

A picture of a typical CTD and associated sensors deployment would be helpful in visualizing the system.

It is an impressive array of in situ equipment that has been employed. How did you decide on which sensors/instruments to use? Perhaps include specific references here to applied deployments by other research groups using the same equipment to demonstrate a broader appeal.

Recognizing that lab round robins were done and these were referenced, can you say a bit more about the HPLC method QA/QC. Extraction efficiencies? Duplicates and RPD? Spikes? Calibration curve information? It seems that in later cruises, the HPLC methodology was changed. This is a natural progression of a longer running research project, but the initial analytical set up and the later set up should be noted in the Methodology.

15.   Lines 427-429 – This passage seems quite critical to the findings, but the wording needs clarification.

16.   Lines 512-515 – these lines don’t seem to fit well here.

17.   Line 547 – The uncertainty is listed as 0.2, but please clarify how the uncertainty is defined/quantified.

18.   Figures and Results:

·       The quality of Figure 1 needs to be improved.

·       Figure 9 – requires additional information/description. For example, I assume the blue dots represent sample sites/sample transects? Please refer to this figure (and all figures) in the text.

·       Figure 10 – there are 3 data points per regression, which is quite a small number and any single value of the 3 may have a large impact. How does this affect the interpretation of your results?

·       Figure 11 (and other following figures) should be placed in the text after it is first referred to.

·       Please discuss and differentiate the results between Figures 10 and 11 in more detail.

·       A deeper discussion of the results for Figure 12 is warranted. For example, there is a much larger anomaly for the results from May 2013. Why would that be the case?

·       Figure 13 – It is unclear where these data points/stations are from, as the x-axis label really does not provide sufficient information.

·       For the 2016 campaign, in what months did it occur and there was only 1 water sample per site?

·       Figure 16 should be a table.

·       Figure 17 – a more informative area map should be included. For a general readership, the study location will be unclear.

·       Figure 20 – similar to my comment regarding Figure 10, above. The regressions have only 3 points and some of the points are quite clustered. How does this affect interpretation of results? Also, should not have a -1 value on the y-axis.

·       The median r2 in Figure 20 may be 0.99, but there is considerable variability in the slopes of the regressions. In this case, r2 may not be the best statistic on which to focus.

·       Figure 21 requires additional information. The individual dots are sample sites?

·       Figures 22 and 23 – it is noted that the r2 increases from 95% to 98%. Is this really a significant increase?

·       Figure 24 – the slope factor color scale should have a title.

·       Figure 25 – this is a critical figure for the paper, but it needs a deeper explanation and discussion. For example, how are the chlorophyll a data linked to the silicates? Exactly what is the influence of the nutrient stock in spatial terms?

19.   In conclusion: a) if you used some of the correction factors determined in your earlier work, by how much would your chlorophyll a estimates be in error and what would the implications be for environmental interpretations?; b) results in Figure 10 show that the sensor slope can be impacted by individual species. Of course in nature, you may get a mix of species, or one species could be dominant. Is this of concern in interpreting the chlorophyll a data for different environments?; and c) So that I am clear, are you recommending that it is not necessary to calibrate sensors with different algae?

20.   I realize this is outside the scope of your study, but does your work also hold some lessons for freshwater monitoring of chlorophyll a?

Author Response

Dear reviewer,

I have finished to correct the manuscript.

Again, let me thank for your pertinent remarks.

Best regards

Reviewer 2 Report

Can you state what is the conclusion?

Author Response

Dear reviewer,

Thank you for your pertinent remarks.

Response to your comment: I added the following paragraph at the end of the paper: “In conclusion, we can say that pre-campaign calibration in algae culture is not recommended. Even with a mixture of algae as close as possible to the one encountered in the field, one cannot predict the physiological state of the cells or the constituents of the sea water that will interfere with the measurement. Only an in situ calibration carried out by comparison with the HPLC analysis of samples taken from the same area allows true values to be obtained.”

Best regards

Reviewer 3 Report

In my review, it was determined that the authors made the specified corrections. Therefore, the article can be accepted in its presentform.

Author Response

Dear reviewer,

Thank you for reading this carefully.

Best regards

Reviewer 4 Report

This topic of this paper is likely to be of interest to Sensors readership.  It summarises a decade of experience in the measurement of in situ chlorophyll concentration.  While not providing any new insights it does provide a technically accurate and scientifically solid piece of work.  

My only comments relate to the presentation that while clear is first, over wordy and would benefit from a careful edit; second, contains some common errors of phraseology.  For example, satellite remote sensing (remote sensing from satellite sensors), calibration (calibration requires one variable to have low or no error), reliability and quality (reliability is a measure of quality), parameter (variable if it varies), guarantee reliability (cannot be done) and qualitative (not a synonym for inaccurate).  Third, uses imprecise colloquiums that detract from the factual nature of the article.  For example, ‘crucial point’, ‘teaches us’, ‘in other words’, ‘better correlation’, ‘in addition to the fact that’, ‘in order’ and ‘answer our problem’.

Author Response

Dear reviewer,

Thank you for reading this carefully. I have taken your remarks into account, clarifying some words and improving my writing style.

Best regards 

Round 2

Author Response

Dear reviewer,

Once again, thank you for allowing me to improve my manuscript!

See my answers in attachment

Kind regards

J. SALAUN
